# Optimization of Outer Diameter Bernoulli Gripper with Cylindrical Nozzle

Roman Mykhailyshyn [1,2,3,*], František Duchoň [4], Ivan Virgala [5], Peter Jan Sinčák [5] and Ann Majewicz Fey [6,7]

1. Texas Robotics, College of Natural Sciences and the Cockrell School of Engineering, The University of Texas at Austin, Austin, TX 78712, USA
2. Department of Automation of Technological Processes and Manufacturing, Ternopil Ivan Puluj National Technical University, 46001 Ternopil, Ukraine
3. EPAM School of Digital Technologies, American University Kyiv, 02000 Kyiv, Ukraine
4. Institute of Robotics and Cybernetics, Slovak University of Technology in Bratislava, 81219 Bratislava, Slovakia; frantisek.duchon@stuba.sk
5. Department of Industrial Automation and Mechatronics, Faculty of Mechanical Engineering, Technical University of Košice, 04200 Kosice, Slovakia; ivan.virgala@tuke.sk (I.V.); peter.jan.sincak@tuke.sk (P.J.S.)
6. Walker Department of Mechanical Engineering, The University of Texas at Austin, Austin, TX 78712, USA; ann.majewiczfey@utexas.edu
7. Department of Surgery, UT Southwestern Medical Center, Dallas, TX 75390, USA
* Correspondence: roman.mykhailyshyn@austin.utexas.edu

**Abstract:** Gripping and manipulating objects using non-contact and low-contact technologies is becoming increasingly necessary in manufacturing. One of the promising contactless gripping technologies is Bernoulli gripping devices for industrial robots. They have many advantages, but when changing the nozzle geometry, it is difficult to find the optimal parameters of the outer diameter of the gripper and its operating parameters. Therefore, the article presents a model for numerical simulation of the dynamics of airflow in the nozzle of the Bernoulli gripping device and in the radial gap between its active surface and the surface of the object of manipulation. Reynolds-averaged Navier–Stokes equations of viscous gas dynamics, SST-model of turbulence, and $\gamma$-model of laminar-turbulent transition were used for this purpose. The technical requirements for the design of the nozzle of Bernoulli jet gripping nozzles are determined and variants of their constructive improvement are offered. According to the results of numerical simulation in the Ansys-CFD software environment, the optimal diameter of the Bernoulli gripping device and the influence of the geometric parameters of the nozzle on the nature of the pressure distribution in the radial gap and its lifting force were determined. Determined the optimal parameters of the height of the gap between the object of manipulation and the Bernoulli gripping device using C—Factor, which will allow efficient operation of Bernoulli gripping devices during automated handling operations using industrial robots.

**Keywords:** robotics; grasping; contactless transportation; Bernoulli gripping device; cylindrical nozzle; CFD

## 1. Introduction

During the implementation of modern automation equipment for handling operations using industrial robots, different types of grippers must be considered [1–6]. For the modern manufacturing of complex systems, it is often necessary to transport objects in a non-contact or low-contact way. The main requirements for the use of such technologies are the characteristics of the object of manipulation (temperature, fragility, flexibility, holes, etc.). Therefore, handling systems of automated production are widely used devices that use the force effect of the jet flowing from the shielded nozzle [7–20]. The interaction between the air jet and the load is influenced by numerous parameters, enabling the utilization of this

interaction's positive effects for various purposes. These purposes include holding cargo through aerodynamic attraction, transporting cargo on an air cushion without contact, and utilizing reactive and viscous friction forces to orient transported objects. These combined interaction effects create a favorable prospect for the development of inventive devices that enable contactless grasping, positioning, and manipulation of industrial facilities.

Currently, the prevailing jet grippers utilize the aerodynamic attraction effect, often referred to as Bernoulli gripping devices (BGD). Given the significance of these grippers, scientists are constantly striving to optimize their design. The initial exploration of the gripper was documented in [21], where the authors elucidated the design by which the gripper generates negative pressure and lifting force. Subsequently, scientists incorporated rubber pads on the gripper's underside to fix the gap between the gripper and the workpiece, ensuring that frictional forces between the pads and the workpiece prevent slippage or falling during horizontal movement [22–28]. Moreover, Ref. [29] devised a contactless gripping device based on the Bernoulli gripper, capable of grasping a silicon wafer.

A key distinction between a Bernoulli gripper and a vacuum cup is that the former releases air externally, creating a negative pressure that prevents air from entering through the inner perimeter. Consequently, even if the workpiece surface is rough, the Bernoulli gripper can maintain negative pressure, empowering it to grip not just sleek objects but also unconventional forms like fabric, textured skin [30–32], and various foods such as meat, tomatoes, and bread [33–35]. BGDs are extensively employed in the radio-electronic industry for manipulating semiconductor plates, solar elements, and printed circuit boards [36,37]. They also find application in the field of printing, specifically with lithographic printing forms in the production process [38].

Prior investigations into Bernoulli grippers have predominantly centered on examining the lifting force in their stable state. In the paper [39], both experimental and theoretical investigations were conducted to examine the dynamic properties of the gripper. In real-world applications, the gripped workpiece is elevated by positioning the gripper directly above it and providing compressed air. During our pick-up experiment, we observed that the workpiece initially oscillated vertically upon lifting, gradually reducing its oscillation amplitude until it reached a stable state. Expanding on this empirical observation, was present a mass-spring-damper model that incorporates the steady-state suction force as a spring, along with an added damping force originating from the flow.

To minimize energy consumption during handling operations with Bernoulli gripping devices (BGD), the authors of article [40] developed a method to optimize gripper orientation during manipulation. Savkiv et al. further present the method for optimizing BGD orientation during transportation operations along both straight and curved trajectories in articles [40,41]. Article [42] presented an adaptive design utilizing Bernoulli gripping devices for grasping metal structures both before and after stapling. One notable aspect of this design is its capability to regulate shape deviations by incorporating a pneumatic sensor into the gripping system. The influence of the input parameters of jet capture devices on their power and energy characteristics is substantiated in the paper [43]. The authors developed a method of designing a cylindrical nozzle for prototyping Bernoulli gripping devices by the 3D printing method [44] and investigated the influence of 3D printing parameters on the power characteristics of the gripping device. Furthermore, the authors in papers [45] address the pressing matter of reducing energy usage in object manipulation. They investigate the energy efficiency of the object manipulation (OM) process by employing an orientation optimization method in various gripping techniques. The economic benefits of utilizing optimal orientation of Bernoulli gripping devices (BGD) during OM are demonstrated by comparing it to transportation without re-orientation. The article [46] investigates the impact of the material center of mass displacement on both the holding force and material movement during the grasping process using a Bernoulli gripping device.

Various methods are employed to optimize and enhance the efficiency of manufacturing and operating gripping devices, depending on their operational principles. Combining

multiple effects is a common approach to achieve improved performance in these devices. In the paper [47], the authors observed a significant enhancement in flow rate characteristics by combining an ejector Coanda effect with a Bernoulli gripper using an annular nozzle. Alternatively, in article [48], a two-stage finite element optimization method was utilized to design a flexible gripper, resulting in maximized friction and retention forces in the three-finger gripper. Topological optimization, aimed at minimizing mass [49–54] and developing cutting-edge micro-grippers [55], is also a crucial stage in the optimization process. To enhance interaction with the manipulated object, optimization of the active surface of the bodies in contact with the object [56–60] or the nozzle elements [61–64] responsible for forming the interacting flow are often employed.

Paper [61] presents computational work that reveals the existence of a power-law relationship between $h_{eq}$ (equilibrium spacing) and the required inlet fluid power to sustain this equilibrium spacing when appropriately scaled. This scaling primarily incorporates the wall shear, while an additional term incorporating the inlet Reynolds number accounts for the applied force on the system. This relationship holds true across various forces, geometric configurations, and material properties. In their paper [62], Shi and Li present a theoretical model that describes the airflow between the gripper and the workpiece. Using this model, they derive theoretical formulas to calculate the pressure distribution and suction force. Their research underscores the substantial influence of the gripper's outer diameter on the suction force, which exhibits a strong correlation with the gap height and the supplied mass flow rate. They examine the connections between the outer diameter and suction force, as well as between the gap height and suction force. Moreover, they present a method for determining the optimal outer diameter based on these relationships. The study also investigates the pressure distribution to explain the effects of varying the outer diameter on the airflow phenomenon. Finally, the authors calculate the values of the optimal outer diameter for different supply mass flow rates and reveal the tendency of the optimal outer diameter to change with varying supply mass flow rates. This information is of significant importance for the design of a Bernoulli gripper. A gripper based on Bernoulli's principle [63] was proposed for handling soft objects with a rough surface and fragile objects. Additionally, in the study, the authors [64] propose a new design of Bernoulli grips to optimize mechanical characteristics. The proposed gripper has the advantages of a larger working area, better anti-interference ability and better impact resistance.

The presented analysis of publications proves that the task of optimizing the design of Bernoulli gripping devices is relevant and appropriate given the expansion of their use in manufacturing processes. In particular, the methods of increasing the energy efficiency of these devices by optimizing their design parameters are insufficiently studied and need further research. Therefore, this paper considers the influence of the rounded shape of the BGD nozzle on its optimal outer diameter. To solve this problem, the approaches of computational hydrodynamics and information technology for the simulation of the finite element method (FEM) are used. FEM allows determining with high accuracy the distributions of pressure, and velocities, to obtain flow lines and other flow parameters.

## 2. Methodology

The key parameters that significantly impact the power and flow characteristics of the Bernoulli gripping device (BGD) (see Figure 1) are as follows: the air pressure in the chamber (labeled as 1), the radius of the nozzle ($r_n$, labeled as 2), the outer radius of the gripper ($r_g = D/2$), and the distance from the edge of the nozzle to the object of manipulation (OM) (labeled as 3). When the gripping device operates with a nozzle radius of $r_n$ and $h_c$ is less than $r_n/2$, the flow experiences the most significant constriction. At this point of maximum constriction, if the gripper is subjected to excessive compressed air pressures exceeding 30 kPa, the flow reaches a critical speed equivalent to the acoustic speed under these conditions. Subsequently, as the radial flow area further expands, its speed becomes supersonic, causing the static pressure on the surface of the OM to decrease below atmospheric levels. At a certain distance from the center of the nozzle, there is a

sudden deceleration of the supersonic flow, transitioning it to a subsonic state, resulting in the formation of a pressure jump. As the flow expands further, the subsonic speed decreases, and the static pressure gradually increases until it reaches atmospheric pressure, marked as $p_a$. The generation of a low-pressure region on the surface of the OM leads to its levitation. Any lateral displacement of the OM is prevented by thrust blocks (labeled as 4).

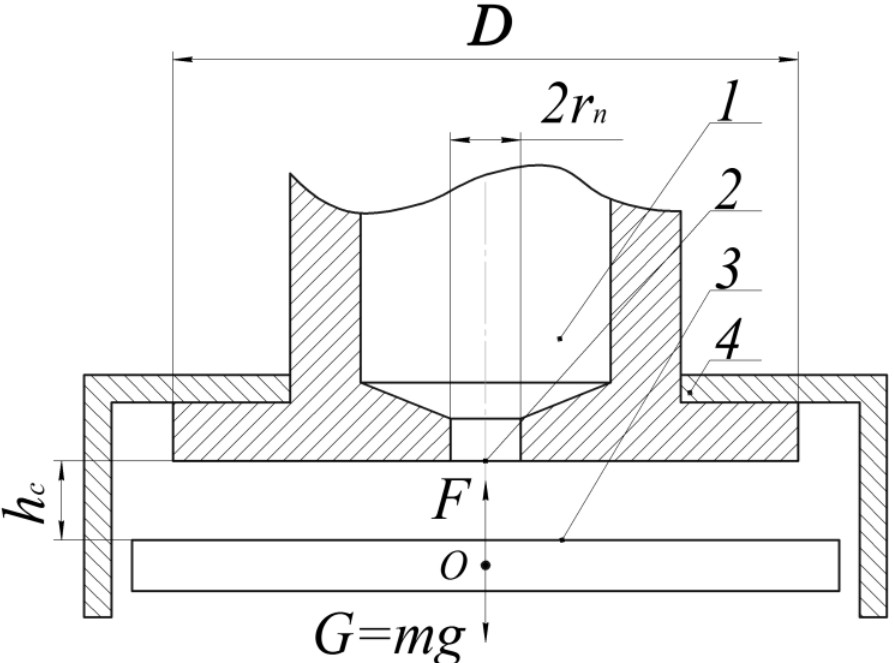

**Figure 1.** Constructive scheme of Bernoulli gripping device.

The mathematical model describing the airflow within the radial interval between the interacting surfaces of the Bernoulli gripping device (BGD) and the object of manipulation (OM) is established using Navier–Stokes equations (specifically, Reynolds-averaged Navier–Stokes equations, RANS) as proposed by Reynolds [65–70]. In order to conduct the modeling process, the SST model of turbulence [71–73] and the $\gamma$-model of laminar and turbulent transition [74–77] are employed.

The $\gamma$-model illustrating the transition from laminar to turbulent flow is characterized by a single differential equation governing the intermittency coefficient $\gamma$:

$$\frac{\partial(\rho\gamma)}{\partial t} + \frac{\partial(\rho V_j \gamma)}{\partial x_j} = P_\gamma - E_\gamma + \frac{\partial}{\partial x_j}\left[\left(\mu + \frac{\mu_t}{\sigma_\gamma}\right)\frac{\partial\gamma}{\partial x_j}\right], \tag{1}$$

where $\rho$—air density; $t$—time; $x$—coordinate; $V$—vector of air velocity; $P_\gamma$, $E_\gamma$—respectively generative and dissipation members of managing directors of laminar and turbulent transition; $\mu$—molecular dynamic viscosity of gas; $\mu_t$—turbulent dynamic viscosity of gas; $\sigma_\gamma = 1.0$—model constant.

The $\gamma$-model of transition employs modified equations derived from the SST model:

$$\frac{\partial}{\partial t}(\rho k) + \frac{\partial}{\partial x_j}(\rho V_j k) = \widetilde{P}_k + P_k^{\text{lim}} - \widetilde{D}_k + \frac{\partial}{\partial x_j}\left((\mu + \sigma_k \mu_t)\frac{\partial k}{\partial x_j}\right), \tag{2}$$

$$\frac{\partial}{\partial t}(\rho\omega) + \frac{\partial}{\partial x_j}(\rho V_j \omega) = \alpha\frac{P_k}{v_t} - D_\omega + Cd_\omega + \frac{\partial}{\partial x_j}\left((\mu + \sigma_\omega \mu_t)\frac{\partial\omega}{\partial x_j}\right), \tag{3}$$

$$\widetilde{P}_k = \gamma P_k, \tag{4}$$

$$\widetilde{D}_k = \max(\gamma, 0.1) \cdot D_k, \tag{5}$$

$$\mu_t = p \frac{a_1 \cdot k}{\max(a_1 \cdot \omega, F_2 \cdot S)}, \tag{6}$$

$$S_{ij} = \frac{1}{2}\left(\frac{\partial V_i}{\partial x_j} + \frac{\partial V_j}{\partial x_i}\right); S^2 = 2S_{ij}S_{ij}, \tag{7}$$

where $k$—kinetic turbulent energy; $\omega$—the specific speed of dissipation of kinetic energy of turbulence; $P_k$, $D_k$—primary formulation responsible for generating and dissipating turbulence; $P_k^{\lim}$—an additional module that ensures the accurate amplification of turbulent viscosity in the transitional region, particularly when the turbulent viscosity of the flowing fluid is extremely low; $\nu_t$—turbulent kinematic viscosity of gas; $\sigma_k$, $\alpha$, $a_1$—empirical constants associated with the model.

The term responsible for generation in Equation (1) takes the form of:

$$P_\gamma = F_{length}\rho S\gamma(1-\gamma)F_{onset}, \tag{8}$$

where, $F_{length}$ represents an empirical correlation that governs the length of the transitional region (taking $F_{length}$ = 100 as a default value); $F_{onset}$ denotes the function that regulates the initiation of transition.

The dissipation component is responsible for the decay or relaminarization process:

$$E_\gamma = c_{a2}\rho\Omega\gamma F_{turb}(c_{e2}\gamma - 1), \tag{9}$$

where $c_{a2}$ and $c_{e2}$ represent empirical constants with values of 0.06 and 50, $\Omega = \sqrt{2\Omega_{i,j}\Omega_{i,j}}$—invariant of the tensor of vorticity; $F_{turb} = e^{-\left(\frac{R_T}{2}\right)^4}$; $R_T = \frac{\rho k}{\mu\omega}$.

The initiation of the laminar and turbulent transition process is regulated by the following functions:

$$F_{onset1} = \frac{Re_v}{2.2Re_{\theta c}}, \ Re_v = \frac{\rho d_\omega^2 S}{\mu}, \tag{10}$$

$$F_{onset2} = \min(F_{onset1}, 2.0), \tag{11}$$

$$F_{onset3} = \max\left(1 - \left(\frac{R_T}{3.5}\right)^3, 0\right), \tag{12}$$

$$F_{onset} = \max(F_{onset2} - F_{onset3}, 0), \tag{13}$$

where $d_\omega$—represents the distance to the nearest wall.

The calculation of the critical Reynolds number for impulsive loss, $Re_{\theta c}$, involves employing an algebraic ratio that incorporates local variables [75]:

$$Re_{\theta c} = f(TU_L, \lambda_{\theta L}). \tag{14}$$

The calculation of $P_k$ generation is performed using the Kato–Launder equation:

$$P_k = \mu_t S\Omega. \tag{15}$$

The supplementary component $P_k^{\lim}$ is defined in the following manner:

$$P_k^{\lim} = 5C_k\max(\gamma - 0.2, 0)(1 - \gamma)F_{on}^{\lim}\max(3C_{SEP}\mu - \mu_t, 0)S\Omega, \\ C_k = 1.0, \ C_{SEP} = 1.0, \tag{16}$$

$$F_{on}^{\lim} = \min\left(\max\left(\frac{\mathrm{Re}_V}{2.2 \cdot \mathrm{Re}_{\theta c}^{\lim}} - 1, 0\right), 3\right), \ \mathrm{Re}_{\theta c}^{\lim} = 1100. \tag{17}$$

### 3. Results and Discussions

The force and operational characteristics of radial flow grippers are typically influenced by the design specifications of the nozzle, active surface, and pneumatic pipeline parameters. A crucial requirement for an optimal design of radial flow grippers is the presence of a smooth, active surface that allows for the seamless expansion of the airflow without any obstructions or disruptions [78,79]. Ensuring a smooth inlet and outlet from the nozzle helps minimize energy losses within the airflow and reduces turbulence forces in the region opposite to the nozzle [43]. To create a rational design of the nozzle element, it is decided to round the nozzle on both sides (Figure 2b) and additionally smooth the transition between the nozzle and the active surface of the gripper due to the chamfer (Figure 2c). Rational designs of Bernoulli gripping devices provide an increase in lifting force, as shown in Figure 2.

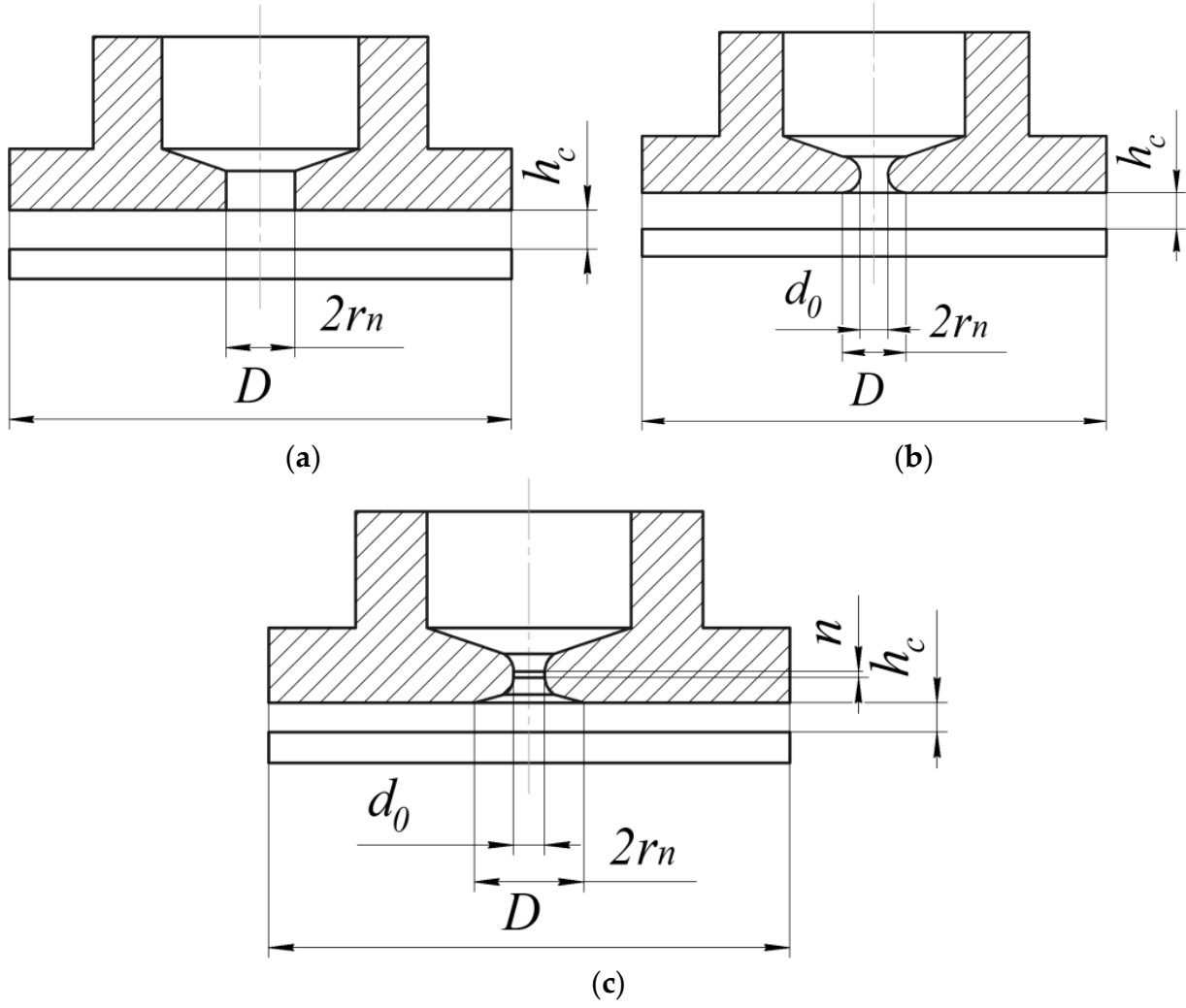

**Figure 2.** Schemes of radial flow grippers with various forms of nozzles: (**a**) classic cylindrical nozzle (**b**) rounded cylindrical nozzle (**c**) rounded cylindrical nozzle with a chamfer.

By incorporating a rounded nozzle (Figure 2b,c), the energy losses experienced by the airflow at the entrance to the radial gap are reduced, allowing for a significant expansion of the airflow. This design approach enables more efficient utilization of the energy within the

airflow, thereby increasing the lifting capability of radial flow grippers when manipulating flat objects.

To minimize the repulsive force when gripping an object of manipulation (with a gap height, $h_0$, greater than 1 mm), it is beneficial for the flow to narrow as it passes through the nozzle and subsequently expand upon entering the radial gap between the active surface of the Bernoulli gripping device (BGD) and the flat object. Such flow geometry is achieved through the design of the BGD with a rounded-off nozzle (Figure 2b,c). The diameter of the smallest section of the rounded-off nozzle should be selected to ensure that the area of this section $\pi d_0^2/4$ is approximately 30% to 50% larger than the area of the critical section, $S^* = 2\pi r_n h_c$.

$$d_0 = \sqrt{(10.4\dots 12)r_n h_c},\tag{18}$$

In order to determine the pressure distribution on the surfaces of the manipulated object, numerical modeling is conducted for the Bernoulli gripping device (BGD) with specific geometric parameters. The BGD used for the modeling has the following dimensions: external diameter ($D$) of 30 mm, nozzle radius ($r_n$) of 3 mm, diameter of the smallest section of the nozzle ($d_0$) of 2.5 mm, height of the radial interval ($h_c$) of 0.2 mm, and neck of the nozzle ($n$) of 0.2 mm.

The numerical modeling is performed using Ansys-CFX, a computational hydro-gas dynamics software. The SST $\gamma$-model of turbulence is utilized for the calculations. To facilitate the calculations, an unstructured grid is constructed in the computational domain using the capabilities of the software, as shown in Figure 3.

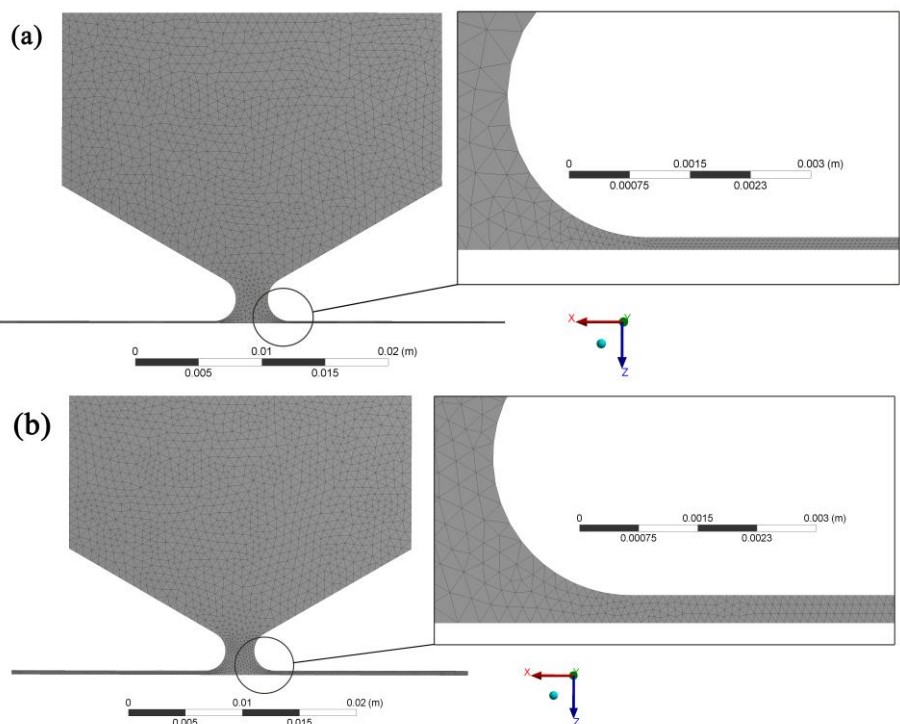

**Figure 3.** Mesh grid of final elements of airflow: (**a**) $h_c$ = 0.15 mm (**b**) $h_c$ = 0.35 mm.

The overall quantity of nodes within the computational domain is dictated by the gripper's outer diameter ($D$) and the gap height ($h_c$) between the object being manipulated and the gripper. To ensure the SST model accurately captures near-wall airflows, it is essential to have a minimum of three elements between the model's walls. This ensures sufficient resolution and accuracy in modeling the boundary layer near the surfaces. The "Num cells across gap" parameter is responsible for the number of such elements in Ansys–CFX–Meshing. With such constraints for different heights $h_c$ (Figure 3), there will be a different number of nodes in the calculation area. As can be seen at a gap height

$h_c = 0.15$ mm (Figure 3a), the number of nodes is 3.2 million and is greater than the number of elements at a gap height $h_c = 0.35$ mm (Figure 3b), which is 1.8 million united in three-dimensional elements (tetrahedra and prisms). The total number of volume elements in the grid is equal to 3–7 million as materials are used, with air as an ideal gas from libraries of the program. Boundary conditions for a model of airflow are presented in Figure 4.

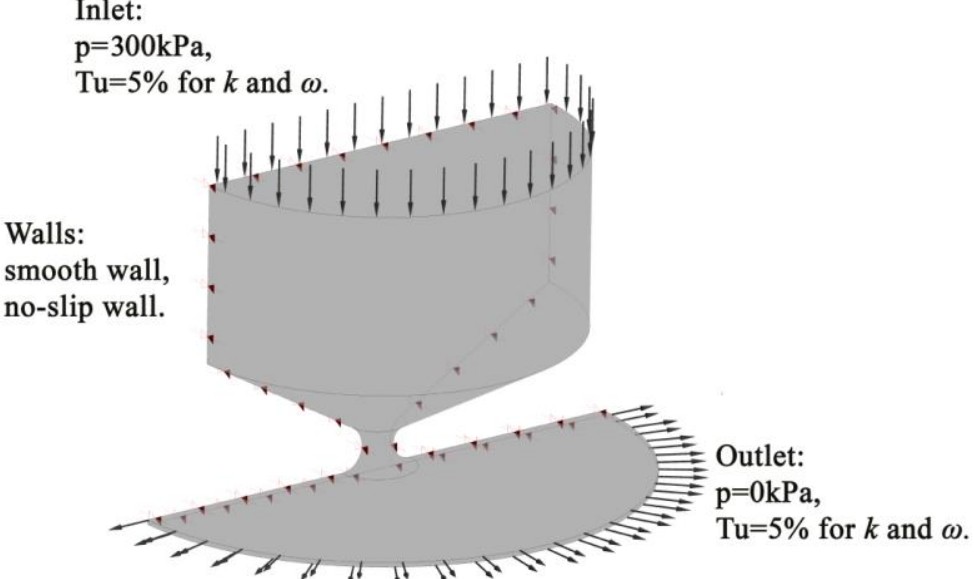

**Figure 4.** Boundary conditions for airflow model.

Based on the results obtained from the calculations using the sonicTurbFoam module, which is specifically designed for turbulent flows of compressed gases at supersonic speeds, pressure distribution profiles on the surfaces of the object of manipulation (OM) have been generated (Figure 5). Additionally, charts depicting the variations in average flow rate within the radial gap have been created (Figure 6).

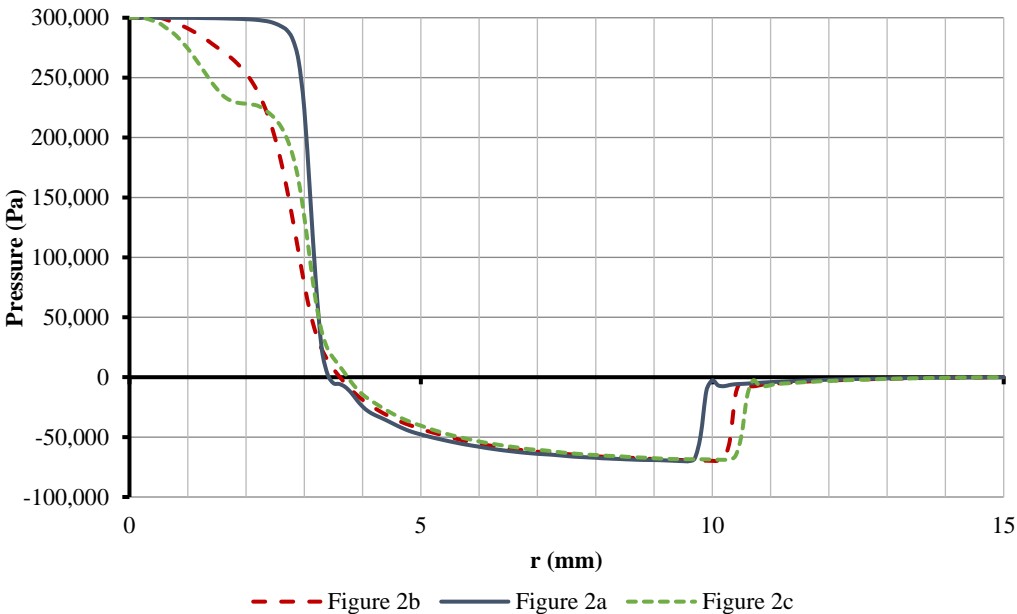

**Figure 5.** Schedules of distribution of pressure upon surfaces of objects of manipulation for various options of design of BGD.

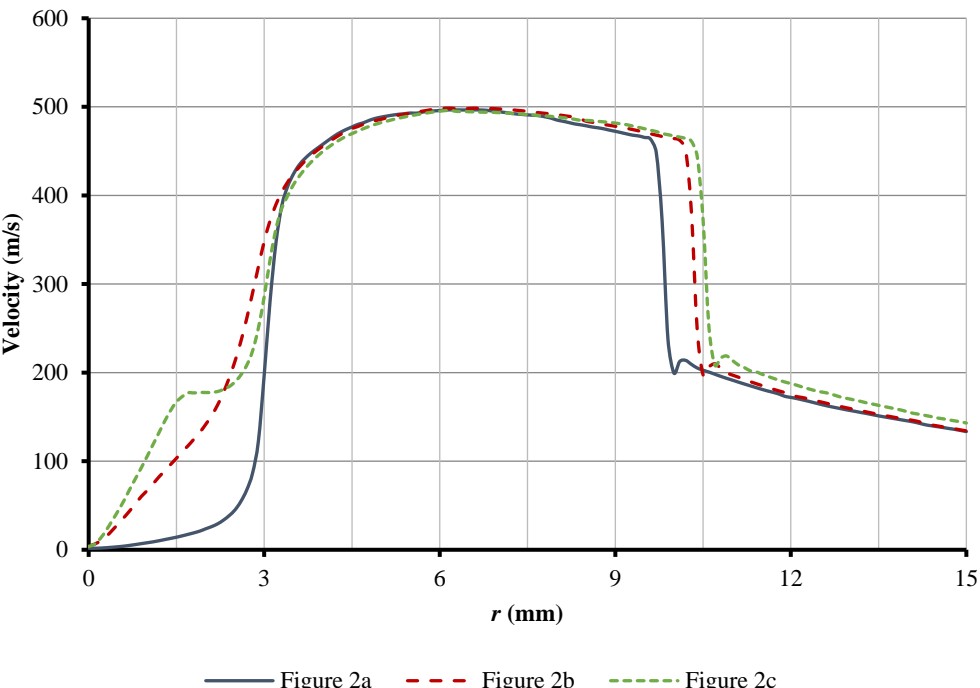

**Figure 6.** Schedules of change of average flow rate in radial gap.

As evident from Figure 5, the utilization of a rounded-off nozzle in the design of the BGD results in a reduction in excessive pressure in the zone opposite the nozzle. In comparison to a BGD with a cylindrical nozzle, the use of a rounded-off nozzle or a rounded-conical nozzle leads to an enlargement of the supersonic zone of depression (Figure 6). This, in turn, generally enhances the lifting force exerted by the gripper on the object of manipulation, which is determined by integrating the distribution of absolute pressure ($p_r$) over the flat surface of the OM:

$$F_l = 2\pi \int\limits_0^{r_g} (p_a - p_r) r dr. \tag{19}$$

Upon integrating the pressure distribution data over the surfaces of the object of manipulation, it has been determined that the upward force is increased by 30–50% for BGD designs utilizing a rounded-off nozzle or a rounded-conical nozzle (Figure 2b,c) compared to the basic design (Figure 2a). Additionally, computational modeling data reveals a compressed air consumption increase of approximately 5–8% when using the rounded-off nozzle and 4–6% when using the rounded-conical nozzle.

Based on studies conducted [43,62] at a constant pressure applied to the Bernoulli gripper, the lifting force $F_l$ will be critically dependent on the height of the gap $h_c$ between OM and BGD. Since the height of the gap $h_c$ is one of the parameters in the theoretical expression of the pressure distribution [10], the pressure distribution between the gripper and OM will depend on the height of the gap $h_c$. To study the effect of $h_c$, the value of the outer diameter of the gripper $D = 30$ mm and the pressure in the gripper chamber $p = 300$ kPa (Figures 7 and 8).

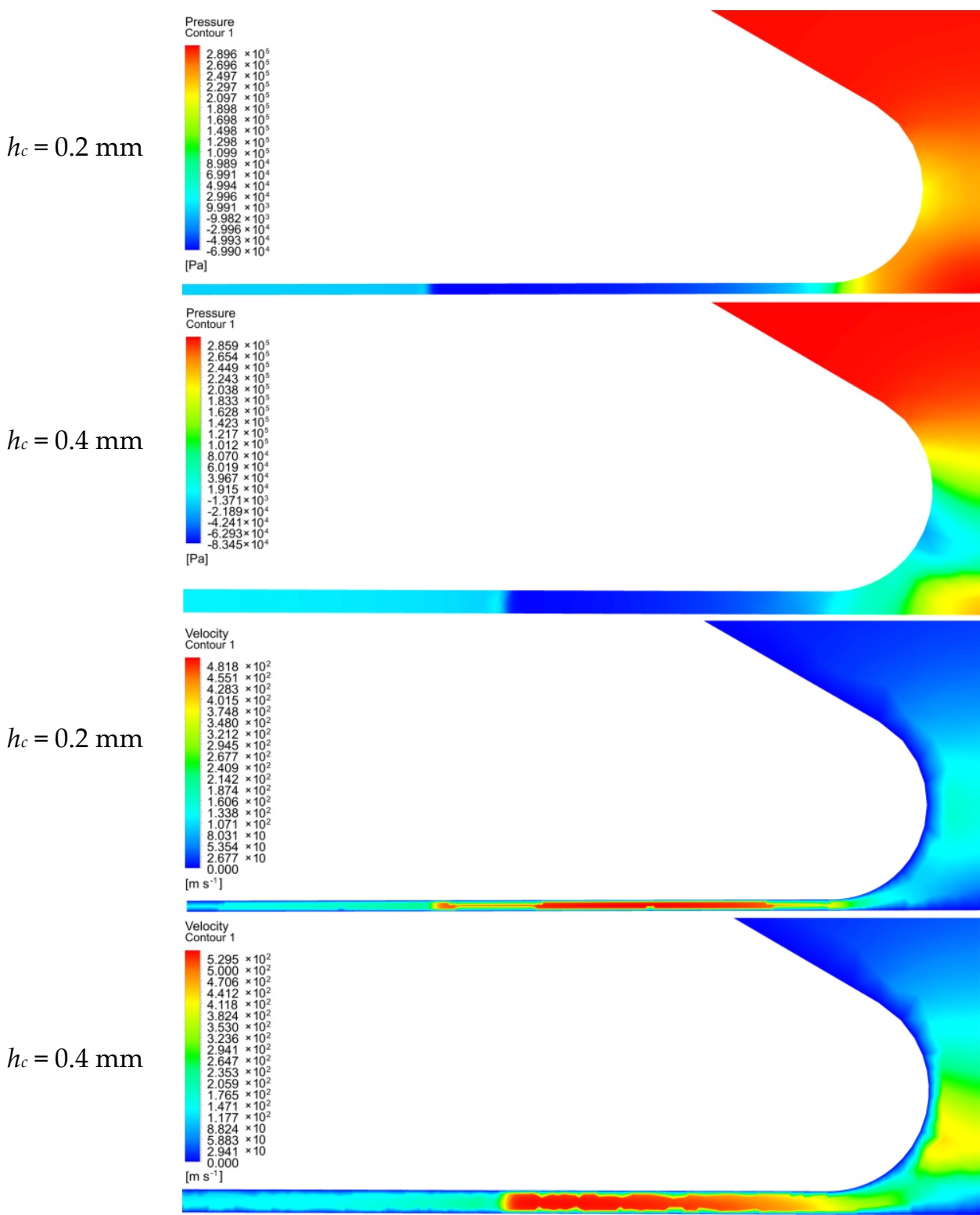

**Figure 7.** The influence of the height of the gap $h_c$ on the velocity and air pressure in the gap between OM and BGD for Figure 2b.

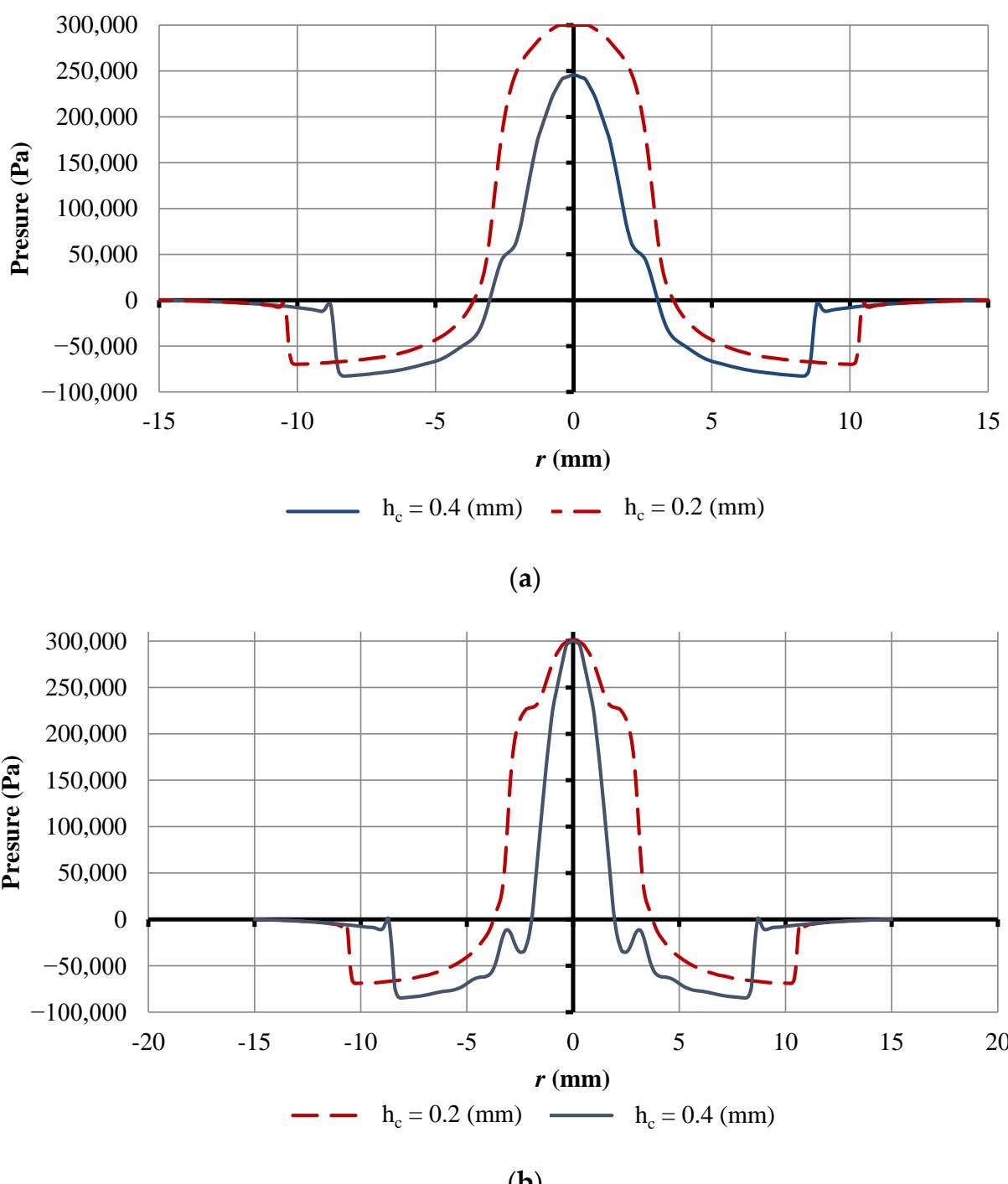

**Figure 8.** The pressure distribution on the surface of the OM at different gap heights $h_c$ and designs: (**a**) Figure 2b; (**b**) Figure 2c.

Figures 7 and 8 show that in the central part in front of the nozzle, there is a high-pressure area because the supply of air coming from the nozzle comes into contact with the part before changing direction. After $r_n$, due to the narrowing of the airflow, the inertial effect of slowing down the air creates a negative pressure in the supersonic velocity of the airflow, while the effect of viscous friction causes a decrease in pressure towards the edge of the gripper. Comparing the curves for different heights of the gap $h_c$, can conclude that:

- In the nozzle area, as the height of the gap increases, the resistance with which the air comes into contact when it enters the gap decreases, as does the speed of air movement. Therefore, the pressure in front of the nozzle decreases with increasing gap height;
- In the area of supersonic air velocity, increasing the height of the gap reduces the effect of narrowing the airflow, which in turn reduces the inertial effect and the effect of viscous friction. This leads to the fact that as the gap increases, the vacuum increases, but the vacuum zone decreases.

The pressure distribution varies depending on the height of the gap as well as the lifting force. The relationship between the lifting force and the height of the gap is the main characteristic of the gripping device. Figure 9 shows the obtained curves, F—h, for two designs of the Bernoulli grippers with a rounded nozzle, which are rounded.

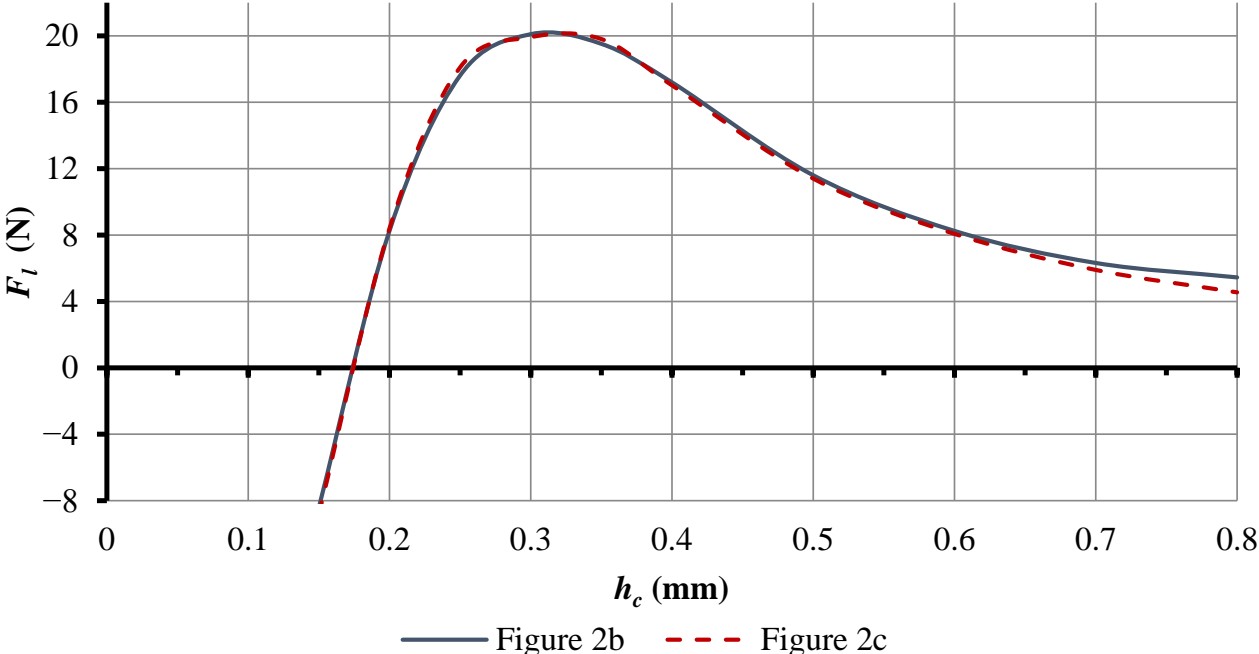

**Figure 9.** Influence of the height of the gap $h_c$ between OM and BGD on the lifting force of BGD of two designs with a rounded nozzle ($D = 50$ mm, $r_n = 3$ mm, $d_0 = 2.5$ mm, $n = 0.2$ mm, $p = 300$ kPa).

When the gap height is small, the lifting force exhibits a negative value, indicating a repulsive force exerted on the object of manipulation. As the gap height increases, the absolute value of the repulsive force diminishes rapidly, approaching zero, and then the lifting force begins to rise. Eventually, the lifting force reaches its peak value before gradually declining. Li and Kagawa extensively discussed this trend in their paper [62]. They demonstrated that the gripper gap exhibits a coexistence of viscous and inertial effects, which stabilize each other. Consequently, when the gap height, $h_c$, is small, the dominant effect is viscous, resulting in the $F_l$—$h_c$ curve tilting in the positive direction. On the other hand, when $h_c$ is sufficiently large, the inertial effect surpasses the viscous effect, leading to a gradual decrease in the lifting force. The transition from the region dominated by the viscous effect to the region dominated by the inertial effect gives rise to a convex curvature in the curve, representing the point of maximum lifting force. The gap height corresponding to the maximum lifting force, $F_l$ max, is referred to as $h_c$ max.

The curve $F_l$—$h_c$ will depend on the change in the outer diameter $D$; Figure 10 shows the $F_l$—$h_c$ curves for grips with a range of outer diameters ($D = 30$, 40, 50, 60, 70, and 80 mm).

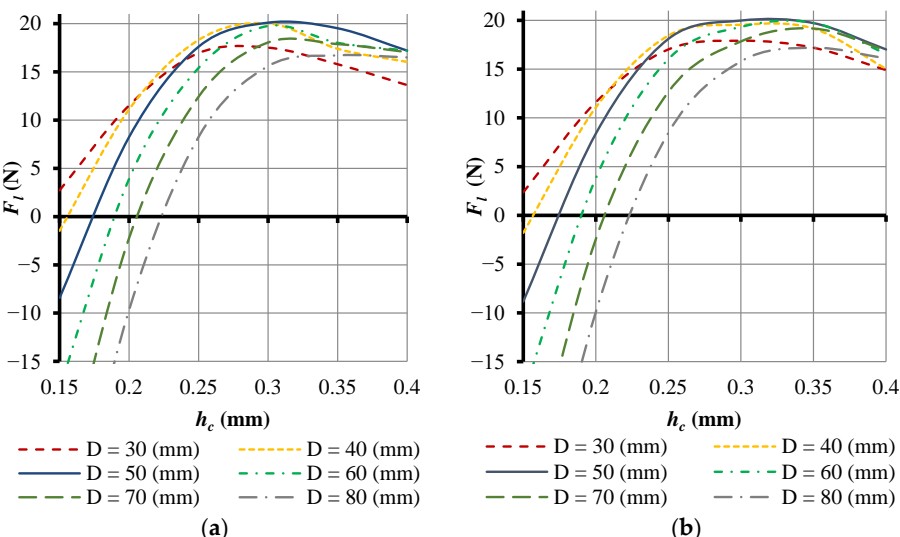

**Figure 10.** Influence of the height of the gap $h_c$ between OM and BGD on the lifting force of BGD at its different diameters ($r_n$ = 3 mm, $d_0$ = 2.5 mm, $n$ = 0.2 mm, $p$ = 300 kPa) and designs: (**a**) Figure 2b; (**b**) Figure 2c.

Despite variations in *D*, all curves exhibit a convex shape, indicating that there exists a specific gap height, $h_c$ max, for each $F_l$—$h_c$ curve that corresponds to the maximum lifting force. By identifying the $h_c$ max values for different outer diameters, the optimal diameter can be determined on the curved $F_l$—$D$—$h_c$ surface, as depicted in Figure 11.

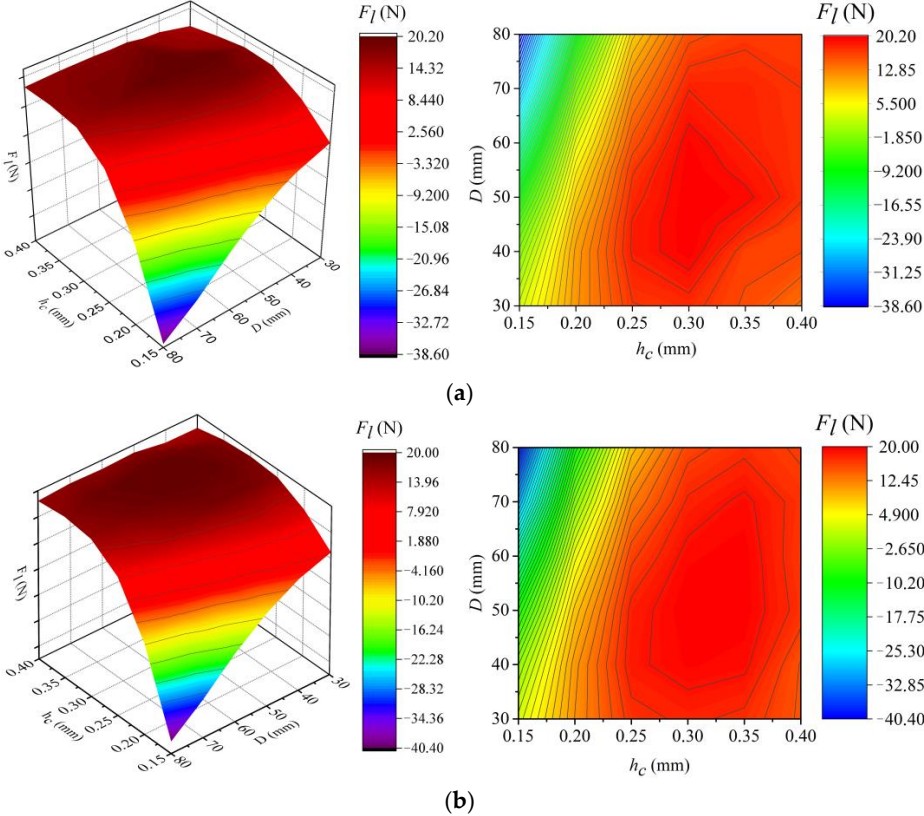

**Figure 11.** $F_l$—$D$—$h_c$ surfaces for two BGD designs ($r_n$ = 3 mm, $d_0$ = 2.5 mm, $n$ = 0.2 mm, $p$ = 300 kPa): (**a**) Figure 2b; (**b**) Figure 2c.

As shown in Figure 11, for both grip designs (Figure 2b; Figure 2c), the lifting force $F_l$ will be maximum in the range of outer diameters $D$ from 40 to 60 mm. For the selected gripping parameters in two cases of the gripping design, at $D$ = 50 mm, the widest range of the maximum lifting force $F_l$ at change $h_c$ is provided.

Since the value of the gap between OM and BGD ($h_c$) has the greatest impact on the gripper rate, and therefore a critical impact on the energy performance of the gripper. To determine the optimum value of the gap between OM and BGD, the so-called C—Factor [80] was selected in Figure 12. As defined by [81], the C-Factor of a gripper can be computed as the ratio of the lifting force it produces over its weight and multiply this ratio by the power. The value obtained is arguably a measure of the efficiency of the gripper and can be used for comparison between different products, designs, etc.

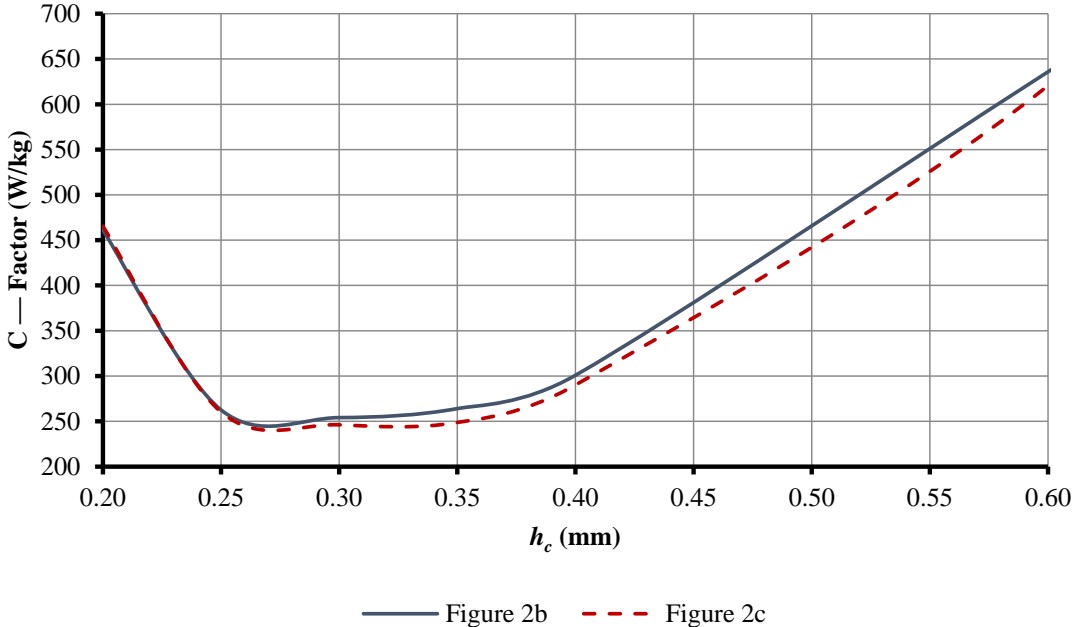

**Figure 12.** Influence of $h_c$ on C—Factor for two BGD designs ($D$ = 50 mm, $r_n$ = 3 mm, $d_0$ = 2.5 mm, $n$ = 0.2 mm, $p$ = 300 kPa).

After analyzing the results of $h_c$ on the C—Factor for two BGD designs, it was found that the optimal gap between OM and BGD was 0.3 mm. This clearance provides maximum lifting force and minimum work spent per kilogram of lifting weight. In addition, of the two designs can be distinguished Figure 2c as the most optimal. This is due to the fact that the design of Figure 2c provides the maximum lifting force in a wider range of diameters $D$ = 40 . . . 70 mm than Figure 2b—$D$ = 40 . . . 60 mm. It is also important that the energy consumption of the gripper of Figure 2, c per weight gain is less than the energy consumption of the gripper of Figure 2, b by 5%. When using such gripping devices when performing automated transport operations in the workplace is quite significant.

## 4. Conclusions

The paper presents a mathematical model for numerically simulating the airflow dynamics in the nozzle and radial gap of a Bernoulli gripping device. Proposed modifications to the nozzle design of industrial robot Bernoulli grippers are suggested. Research findings indicate that the diameter of the smallest section of the rounded nozzle should be chosen to be 30 . . . 50% larger than the critical section area.

The use of a rounded nozzle in the Bernoulli gripping device design reduces excess pressure on the manipulated object's surface opposite the nozzle. This improvement enhances the power characteristics of Bernoulli grippers by 40 . . . 50% while only increasing compressed air consumption by 5 . . . 9%. The outer diameter of the gripper, $D$, signif-

icantly influences the lifting force and is closely related to the gap height, *h*. Thus, the relationship between them is described using the curvilinear surface *F—D—h*. Optimal outer diameter parameters are determined to minimize the viscosity's weakening effect on negative pressure, resulting in maximum lifting force. For a nozzle with a diameter of 6 mm, the optimal Bernoulli gripping device diameter is determined to be 50 mm.

The C-Factor was used to determine the optimal value of the gap height between the object to be manipulated and the Bernoulli gripping device. For a nozzle with a diameter of 6 mm, the optimal parameter of the gap height of 0.3 mm and the rational design of the nozzle of the Bernoulli gripping device are determined.

Given the primary benefit of jet grasping systems lies in their ability to minimize object damage and contact, future research will focus on developing grasping systems specifically for medical applications. These systems aim to prevent tissue damage and minimize contact, building upon the existing research conducted in this paper.

**Author Contributions:** Conceptualization, R.M. and A.M.F.; methodology, R.M. and F.D.; software, I.V. and A.M.F.; formal analysis, R.M. and P.J.S.; investigation, R.M. and A.M.F.; resources, R.M.; writing—original draft preparation, R.M. and A.M.F.; writing—review and editing, R.M., F.D. and I.V.; visualization, R.M. and P.J.S.; project administration, A.M.F.; funding acquisition, R.M. and F.D. All authors have read and agreed to the published version of the manuscript.

**Funding:** This article was written thanks to the generous support under the Operational Program Integrated Infrastructure for the project: "Research and the development of the applicability of autonomous flying vehicles in the fight against the pandemic caused by COVID-19", Project no. 313011ATR9, co-financed by the European Regional Development Fund.

**Institutional Review Board Statement:** Not applicable.

**Informed Consent Statement:** Not applicable.

**Data Availability Statement:** Not applicable.

**Acknowledgments:** The authors would like to thank the Slovak Grant Agency—project KEGA 028STU-4/2022 and VEGA 1/0436/22 "Research on modeling methods and control algorithms of kinematically redundant mechanisms".

**Conflicts of Interest:** The authors declare no conflict of interest.

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
