# Peer review of "Optimization of Outer Diameter Bernoulli Gripper with Cylindrical Nozzle"

_machines, doi:10.3390/machines11060667_

Round 1

Reviewer 1 Report

The authors focus on a mathematical model for numerically simulating the airflow dynamics in the nozzle and radial gap of a Bernoulli gripping device. The manuscript needs further improvement according to the following comments and suggestions.

1. Theoretical solutions can be provided to verify the effectiveness of the established simulation model.

2. In microobject manipulation, It is necessary to avoid the surface damage caused by the lifting force. How can you ensure the nondestructive gripping?

Minor editing of English language required。

Author Response

We appreciate your consideration and time devoted to reviewing our manuscript, which provided careful comments to improve the manuscript.

1. Theoretical solutions can be provided to verify the effectiveness of the established simulation model.

Thank you very much for your comment, the theoretical model has already been presented in the paper [1] and is part of a series of articles on this topic [2-3].

2. In microobject manipulation, It is necessary to avoid the surface damage caused by the lifting force. How can you ensure the nondestructive gripping?

Grasping fragile and easily deformable objects with minimal contact is the main task of most manufacturing. It is a jet gripping devices that have the ability to minimize contact with the object due to non-contact grasping. This article focuses on one of several possible designs for the Bernoulli Jet Gripping Device [4]. Due to the good dynamic characteristics of the Bernoulli gripper, when the object is lifted to the gripper, a pneumatic cushion is formed in the gap between the object and the gripper, which prevents contact between them [5]. Therefore, in the absence of friction elements, contact does not occur, only side stops are used to protect the object from slipping. In the case of the presence of friction elements, the contact force can be adjusted due to the height of the friction elements and the supply pressure.

We are currently conducting research on the use of low-contact jet grippers for medical applications to minimize tissue damage. Therefore, we will develop this direction in the following works.

References

[1]    Savkiv, V.; Mykhailyshyn, R.; Duchon, F. Gasdynamic analysis of the Bernoulli grippers interaction with the surface of flat objects with displacement of the center of mass. Vacuum 2019, 159, 524-533. https://doi.org/10.1016/j.vacuum.2018.11.005

[2] Mykhailyshyn, R.; Duchoň, F.; Mykhailyshyn, M.; Majewicz Fey, A. Three-Dimensional Printing of Cylindrical Nozzle Elements of Bernoulli Gripping Devices for Industrial Robots. Robotics 2022, 11, 140. https://www.mdpi.com/2218-6581/11/6/140

[3]    Mykhailyshyn, R.; Xiao, J. Influence of inlet parameters on power characteristics of Bernoulli gripping devices for industrial robots. Applied Sciences 2022, 12(14), 7074. https://doi.org/10.3390/app12147074

[4]    Mykhailyshyn, R.; Savkiv, V.; Maruschak, P.; Xiao, J. A systematic review on pneumatic gripping devices for industrial robots. Transport 2022, 37(3), 201-231. https://doi.org/10.3846/transport.2022.17110

[5]    Mykhailyshyn, R.; Savkiv, V.; Duchon, F.; Chovanec, L. Experimental Investigations of the Dynamics of Contactless Transportation by Bernoulli Grippers. In Proceedings of the 2020 IEEE 6th International Conference on Methods and Systems of Navigation and Motion Control (MSNMC), Kyiv, Ukraine, 20–23 October 2020; pp. 97–100.

Reviewer 2 Report

The “Introduction” is organized, with precise references to what has been achieved until now in the field. I really liked the organization of the Introduction.

I consider that few issues should be clarified, namely:

1. In the chapter "Methodology" theoretical considerations and 17 equations are presented. In the chapter "Results and discussions" there are no references to the 17 equations and so it is not very clear how they were used. This makes it difficult to reproduce the proposed model in another research laboratory.

2. The proposed novelty element of the paper consists of the new geometry of nozzle. However, concrete details regarding this geometry (radii of curvature, etc.) are missing. There is also a lack of considerations in relation to which the synthesis of this geometry has been done.

3. The number of articles cited and published in "Machines" is too small and I recommend increasing it.

In conclusion I consider that the paper can be improved in relation to the observations.

Author Response

We appreciate your consideration and time devoted to reviewing our manuscript, which provided careful comments to improve the manuscript.

1. In the chapter "Methodology" theoretical considerations and 17 equations are presented. In the chapter "Results and discussions" there are no references to the 17 equations and so it is not very clear how they were used. This makes it difficult to reproduce the proposed model in another research laboratory.

The mathematical model presented in the Methodology section provides a mathematical generalization of the SST finite element method used for modeling in Ansys. However, in order for other laboratories to be able to conduct the same research using other software such as Abacus, Сomsol and develop their own, this mathematical model is provided.

2. The proposed novelty element of the paper consists of the new geometry of the nozzle. However, concrete details regarding this geometry (radii of curvature, etc.) are missing. There is also a lack of considerations in relation to which the synthesis of this geometry has been done.

Added to the text (lines 204-207):
To create a rational design of the nozzle element, it is decided to round the nozzle on both sides (Figure 2, b), and additionally smooth the transition between the nozzle and the active surface of the gripper due to the chamfer (Figure 2, c). 

Already in the text (lines 219-228):
The diameter of the smallest section of the rounded-off nozzle should be selected to ensure that the area of this section is approximately 30% to 50% larger than the area of the critical section, S*=2πrnhc.
equation(18)
In order to determine the pressure distribution on the surfaces of the manipulated object, numerical modeling is conducted for the Bernoulli gripping device (BGD) with specific geometric parameters. The BGD used for the modeling has the following di-mensions: external diameter (D) of 30 mm, nozzle radius (rn) of 3 mm, diameter of the smallest section of the nozzle (d0) of 2.5 mm, height of the radial interval (hc) of 0.2 mm, and neck of the nozzle (n) of 0.2 mm.

3. The number of articles cited and published in "Machines" is too small and I recommend increasing it.

Reference list updated.

In conclusion, I consider that the paper can be improved in relation to the observations.

Added to the text:
Given the primary benefit of jet grasping systems lies in their ability to minimize object damage and contact, future research will focus on developing grasping systems specifically for medical applications. These systems aim to prevent tissue damage and minimize contact, building upon the existing research conducted in this paper.